# Giant voltage-controlled modulation of spin Hall nano-oscillator damping

Himanshu Fulara [1✉], Mohammad Zahedinejad[1,2], Roman Khymyn [1], Mykola Dvornik[1,2], Shunsuke Fukami [3,4,5,6,7], Shun Kanai [3,4], Hideo Ohno[3,4,5,6,7] & Johan Åkerman [1,2,8✉]

Spin Hall nano-oscillators (SHNOs) are emerging spintronic devices for microwave signal generation and oscillator-based neuromorphic computing combining nano-scale footprint, fast and ultra-wide microwave frequency tunability, CMOS compatibility, and strong non-linear properties providing robust large-scale mutual synchronization in chains and two-dimensional arrays. While SHNOs can be tuned via magnetic fields and the drive current, neither approach is conducive to individual SHNO control in large arrays. Here, we demonstrate electrically gated W/CoFeB/MgO nano-constrictions in which the voltage-dependent perpendicular magnetic anisotropy tunes the frequency and, thanks to nano-constriction geometry, drastically modifies the spin-wave localization in the constriction region resulting in a giant 42% variation of the effective damping over four volts. As a consequence, the SHNO threshold current can be strongly tuned. Our demonstration adds key functionality to nano-constriction SHNOs and paves the way for energy-efficient control of individual oscillators in SHNO chains and arrays for neuromorphic computing.

[1] Physics Department, University of Gothenburg, 412 96 Gothenburg, Sweden. [2] NanOsc AB, Electrum 229, 164 40 Kista, Sweden. [3] Laboratory for Nanoelectronics and Spintronics, Research Institute of Electrical Communication, Tohoku University, 2-1-1 Katahira, Aoba-ku, Sendai 980-8577, Japan. [4] Center for Spintronics Research Network, Tohoku University, 2-1-1 Katahira, Aoba-ku, Sendai 980-8577, Japan. [5] Center for Innovative Integrated Electronic Systems, Tohoku University, 468-1 Aramaki Aza Aoba, Aoba-ku, Sendai 980-0845, Japan. [6] Center for Science and Innovation in Spintronics, Tohoku University, 2-1-1 Katahira, Aoba-ku, Sendai 980-8577, Japan. [7] WPI-Advanced Institute for Materials Research, Tohoku University, 2-1-1 Katahira, Aoba-ku, Sendai 980-8577, Japan. [8] Material and Nanophysics, School of Engineering Sciences, KTH Royal Institute of Technology, Electrum 229, 164 40 Kista, Sweden. ✉email: himanshu.fulara@physics.gu.se; johan.akerman@physics.gu.se

High-density arrays of mutually coupled nanoscale spintronic oscillators have the potential to closely mimic the behavior of the non-linear oscillatory neural networks of the human brain and thus open the way for efficient high-speed and low-power neuromorphic computing and signal processing[1–3]. Recently, a proof-of-principle demonstration of vowel recognition using a chain of four electrically coupled spin-torque nano-oscillators (STNOs) with individual drive current control was reported[4]. However, the current-based tunability is neither an energy-efficient approach nor an ideal arrangement to scale the neuromorphic computing to large dynamical neural networks.

Spin Hall nano-oscillators (SHNOs)[5–9] have recently shown enormous potential as an energy-efficient alternative to conventional STNOs thanks to their easy fabrication process, flexible device geometry, reduced power dissipation, and CMOS compatibility[10]. SHNOs take advantage of a pure spin current, produced by the spin Hall effect in a non-magnetic heavy-metal layer, to excite the local magnetization of a magnetic thin film into steady-state auto-oscillating precession at microwave frequencies. Among a variety of SHNO device layouts presented so far, nano-constriction based SHNOs[10–14] exhibit robust mutual synchronization both in linear chains[15] and in two-dimensional arrays of as many as 64 SHNOs[16]. 4 × 4 arrays of 16 such SHNOs were also used to demonstrate[16] the same type of synchronization maps as in ref. [4] employed for neuromorphic vowel recognition. However, a major challenge that has to be addressed, before more complex neuromorphic tasks can be attempted, is the individual frequency control of each SHNO inside the network. It is also of particular interest to be able to turn individual networked SHNOs on and off at will as this forms the basis of other recent oscillator network-based computing paradigms[17,18].

Voltage-controlled magnetic anisotropy (VCMA)[19–29] has recently attracted great interest as it offers a highly energy-efficient approach to drive magnetization switching with significantly lower energy dissipation compared to current control in, e.g., spin-transfer torque-MRAM[22,23,30]. Here, we demonstrate how VCMA can also provide strong individual control of the threshold current and the auto-oscillation frequency of nano-constriction based $W(5 nm)/(Co_{0.75}Fe_{0.25})_{75}B_{25}(1.7 nm)/MgO(2 nm)/AlO_x(2 nm)$ SHNOs with relatively strong interfacial perpendicular magnetic anisotropy (PMA)[9]. We observe a giant 42% modulation of the total effective damping over four volts (10.5%/V), a very large 22% modulation in threshold current (5.5%/V), and a robust 50 MHz frequency tunability (12 MHz/V). Our detailed analysis, using spin-torque ferromagnetic resonance (ST-FMR) and micromagnetic simulations, shows that these large effects are caused by a moderate VCMA, which tunes the auto-oscillation frequency of the nano-constriction such that its mode volume and coupling to propagating spin waves in the extended CoFeB layer can be greatly varied. When the coupling to the surrounding spin waves increases, the auto-oscillating mode experiences an increasing load due to radiation of magnons into the magnetic leads, which is experimentally observed as strongly increased effective damping in the ST-FMR measurements and an increased threshold current of the SHNOs. The increased frequency tunability and the possibility to turn individual oscillators on/off within a large network will allow for recently suggested oscillator computing approaches to be implemented using voltage-controlled SHNOs.

## Results

### Nano-patterned gated SHNO device schematic and layout with individual layers. 

Figure 1a shows the optical microscope image of an electrically gated nano-constriction SHNO device with electrical signal pads and how these connect to the experimental set-up. The top signal-ground pads are utilized to apply different gate voltages via the gate electrode. The bottom signal-ground pads are used to locally inject the electrical drive current into the device and to pick up the microwave signals from the spin-orbit torque (SOT) driven auto-oscillations of the CoFeB magnetization. Figure 1b shows an enlarged zoomed-in area of the gated nano-constriction SHNO; a further zoomed-in scanning electron microscope (SEM) image of the nano-constriction is displayed in Fig. 1c. A positive direct current (d.c.) is injected along the $y$-direction while $\phi$ and $\theta$ define the in-plane (IP) and out-of-plane (OOP) field angles, respectively, for the applied magnetic field. The two different schematic cross-sectional views of the material stack with structure $\beta\text{-}W(5 nm)/(Co_{0.75}Fe_{0.25})_{75}B_{25}(1.7 nm)/MgO(2 nm)/AlO_x(2 nm)$ grown on highly resistive Si substrates are shown in Fig. 1d, e. The $\beta$-phase of W is known to produce large SOT[13,31,32] and the thinner CoFeB exposed to MgO layer facilitates the PMA in the CoFeB layer[33].

### Voltage-controlled tunabilities of threshold operational current and auto-oscillation frequency. 

Figure 2a–c shows color plots of the current-dependent auto-oscillation power spectral density (PSD) excited at a fixed OOP field strength of 0.4 T and subject to three different gate voltages. Note that the leakage current does not exceed 5 nA at $V_G = \pm 2 V$ during auto-oscillation measurements. One can observe a dramatic reduction of the threshold current ($I_{th}$) as the gate voltage increases from −2 to +2 V. We extract $I_{th}$ via a linear fit of $1/P$ vs. $I_{dc}$ (employing the method described in ref. [34]), as shown in the inset of Fig. 2d for one representative gate voltage, $V_G = -1 V$. The main panel of Fig. 2d displays the variation of $I_{th}$ as a function of gate voltage exhibiting a linear dependence with a very large overall modulation of 22% with $V_G = \pm 2 V$. In Fig. 2e, we show a comparison of the auto-oscillation frequencies at different gate voltages, exhibiting a remarkable 50 MHz frequency tunability with gate voltage. We analyze the tunability behavior by extracting the auto-oscillation frequency shift $\Delta f(V_G) = f(V_G) - f(V_G = 0)$ at 1.2 mA, exhibiting a linear dependence on gate voltage with a rate of 12 MHz/V, as shown in Fig. 2f. We note that this is twice as large as the frequency tunability demonstrated in nano-gap SHNOs using back-gated permalloy films[35], corroborating the benefit of VCMA in CoFeB/MgO-based system for frequency tuning.

### Spin-torque ferromagnetic resonance measurements on a gated SHNO device. 

To understand the origin of the voltage-controlled tunability of threshold current and auto-oscillation frequency, we performed detailed ST-FMR measurements on a similar gated 150 nm nano-constriction SHNO device subject to different microwave frequencies ranging from 4 to 12 GHz at different gate voltages. In Fig. 3a, we show typical ST-FMR peaks recorded at a frequency of 7 GHz for six different gate voltages starting from $V_G = +1 V$ to $V_G = -3 V$. The mixing voltages $V_{mix}$ are offset along the $y$-axis to enable comparison of the line shapes. Figure 3b shows the variation of the effective magnetization $\mu_0 M_{eff}$, extracted from a Kittel fit (Eq. (1)) of field-dependent resonance peak positions at different microwave frequencies ranging from 4 to 12 GHz, as a function of gate voltage. We observe a linear dependence of $\mu_0 M_{eff}$ with gate voltage, exhibiting an overall 14% modulation under electrostatic gating of $\Delta V_G = 4 V$. Using equation $H_k^\perp = M_S - M_{eff}$ with the saturation magnetization, $\mu_0 M_S = 1.3 T$, this translates into a minor tunability (<1%) of the interfacial PMA. Figure 3c displays the plot of line-widths extracted as half-width at half maximum (HWHMs) from the ST-FMR peaks as a function of different microwave

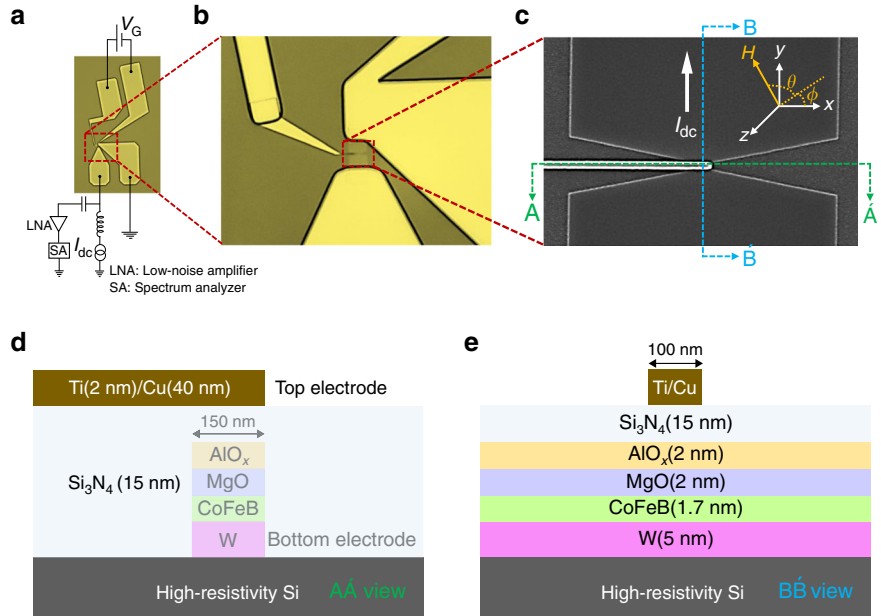

**Fig. 1 Gated SHNO schematic and material stack. a** Optical microscope image of a gated SHNO and how it connects to the experimental set-up. **b** Enlarged zoomed-in optical image showing the gated SHNO. **c** A scanning electron microscope (SEM) image of a gated SHNO of width 150 nm. Schematic of the material stack showing layer order and thicknesses along **d** the gated electrode, and **e** the current direction.

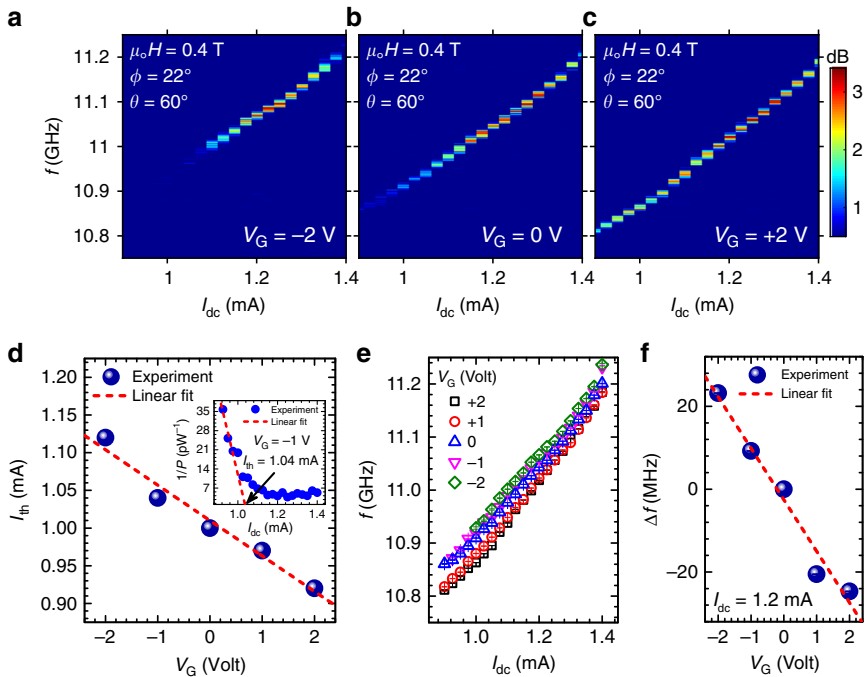

**Fig. 2 Voltage-controlled threshold current and auto-oscillation frequency.** PSDs of the spin-wave auto-oscillations vs. current showing a systematic change both in threshold current ($I_{th}$) and in frequency under the application of different gate voltages **a** $V_G = -2$ V, **b** $V_G = 0$ V, and **c** $V_G = +2$ V. **d** Variation of $I_{th}$ with gate voltage exhibiting a linear dependence. Inset: Illustration of threshold current extraction via a linear fit of $1/P$ vs. $I_{dc}$ (employing the method as in ref. [34]). **e** Comparison of auto-oscillation frequency tunability with drive current at different gate voltages. **f** Dependence of auto-oscillating frequency shift, $\Delta f(V_G) = f(V_G) - f(V_G = 0)$, on gate voltage at a drive current $I_{dc} = 1.2$ mA well above threshold.

frequencies under different gate voltages. The experimental data together with their respective linear fits to $\Delta H = \Delta H_0 + 2\pi\alpha f/\gamma\mu_0$ (solid lines) with a gyromagnetic ratio of $\gamma/2\pi = 29.1$ GHz/T show a remarkable variation in line-widths indicating a dramatically large modulation in effective damping constant, $\alpha$. In Fig. 3d, we show that the modulation of the effective damping

constant as a function of gate voltage is as large as 42%, exhibiting a linear dependence with a slope of 0.002/V.

Note that our ST-FMR measurements are sensitive only to the area in the vicinity of the gate, since ST-FMR peak shifts with the applied gate voltage. Such a result can be somewhat expected since the applied current density is only substantial in

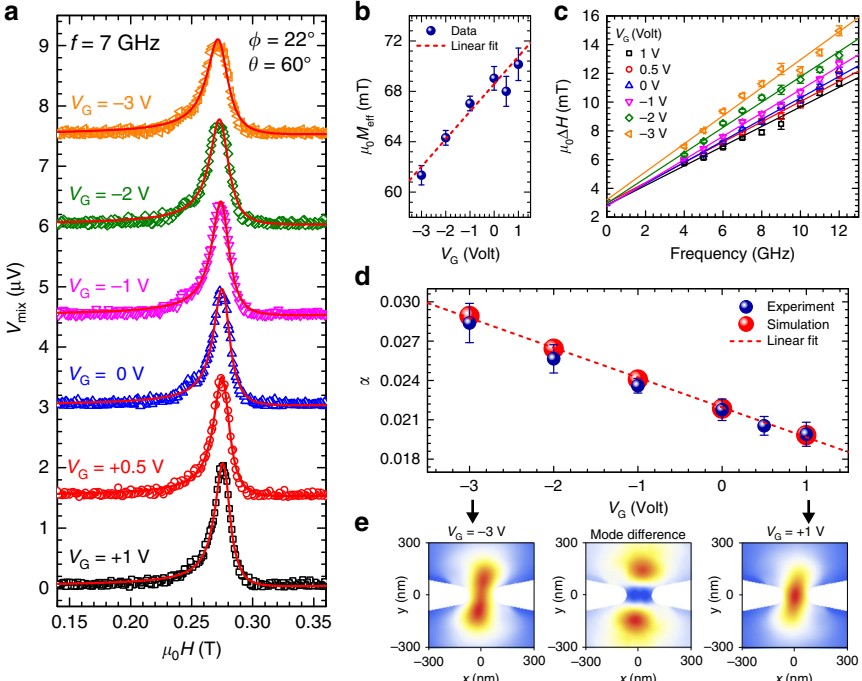

**Fig. 3 Voltage-controlled effective magnetization, line-width, and damping constant. a** Typical ST-FMR spectra recorded at a frequency of 7 GHz for different gate voltages. The red solid lines are fits to a sum of symmetric and anti-symmetric Lorentzian functions. **b** Gate voltage dependence of the effective magnetization $\mu_0 M_{eff}$ extracted from Kittel fits (Eq. (1)) to ST-FMR data in the thin-film approximation. **c** Extracted line-width (HWHM) vs. resonance frequency for each gate voltage along with linear fits shown by solid lines. **d** Variation of the effective damping constant $\alpha$ with gate voltage obtained from the experiment (blue) and simulation (red) displaying a strong linear dependence. **e** Spatial distribution of the excited linear modes normalized by their maximum amplitude displaying stronger localization at $V_G = +1$ V (right panel) relative to the one at $V_G = -3$ V (left panel). The middle panel shows their corresponding amplitude difference. All error bars indicate the standard error obtained from fitting the experimental data.

the nano-constriction region, i.e., under the gate. In addition, the strongly non-uniform nature of the spin-current density does not allow to excite the magnetization uniformly over the whole sample[36]. To gain deeper insight into the observed effective damping modulation, we carried out micromagnetic simulations to study the spatial amplitude profiles of the dynamical mode in the nano-constriction region under different gate voltages for a 150 nm SHNO.

Our simulations show that for the given device configuration with the large value of PMA, the demagnetizing field does not create substantial confinement for the magnons in the constriction region, which means the absence of complete localization of the linear-mode[9]. The mode consists of an oscillating core in the center, coupled with the magnons propagating out of the nano-constriction. The total energy loss of the excited mode hence consists of both the intrinsic damping of the core and the energy radiated away by the emitted magnons[37]. The latter is quite substantial in our case, due to the small volume of the excited core.

The applied voltage on the gate, through the above-mentioned tunability of the interfacial PMA, changes the potential for the magnons in the constriction region, which in turn changes the ratio of magnons in the core and emitted spin waves[6,37]. Thus, the negative voltage increases the PMA value, which raises the potential and leads to the further delocalization of the mode and, in turn, a larger value of the emission losses and total damping via the relaxation rate of the excited dynamical mode. On the other hand, a positive voltage leads to a more localized behavior of the excited mode with predominantly intrinsic losses of the core, which substantially reduces the total damping. The spatial profiles of the mode amplitudes $|\tilde{m}_x|$ normalized to their respective maximum amplitudes are shown in Fig. 3e for $V_G = -3$ V and

$V_G = +1$ V. Note that the core of the mode in the center of the nano-constriction is substantially more pronounced for $V_G = +1$ V whereas less localized for $V_G = -3$ V. The extracted values of the total damping from simulation, as shown in Fig. 3d by red spheres with a linear fit, are in a good agreement with the experimental results. While our simulations reproduce quite well the experimentally observed effective damping modulation, we do not rule out the possibility of a minor change in intrinsic damping under the application of gate voltage[25].

We also performed ST-FMR measurements at different levels of a device current under positive field direction at two fixed microwave frequencies of 6 and 12 GHz, respectively. The variations of extracted ST-FMR linewidth as a function of dc current at different gate voltages for the reference SHNO device are shown in Fig. 4a, b. We fit the linewidth variations to a linear function and the extracted slopes of lines are plotted as a function of gate voltages for both frequencies in Fig. 4c, d. Each value of the slope determines the efficiency of the linewidth change with dc current and is, therefore, a measure of the extent to which the excited mode is affected by the drive current. As the W/CoFeB interface is unlikely to get modified by applied gate voltage due to effective screening of electric field within the CoFeB(1.7 nm) layer, the impact of gating on the spin Hall efficiency (SHE) should be minimal and the slope, therefore, depends on the volume of the excited mode[37]. The observed variation of the slope with the gate voltage is also in a good agreement with our simulation results exhibiting stronger localization of the dynamical mode at +1 V and weaker localization at −3 V, as shown in Fig. 3e. This is because the strongly localized mode resides in the nano-constriction region and has a greater overlap with the current density profile.

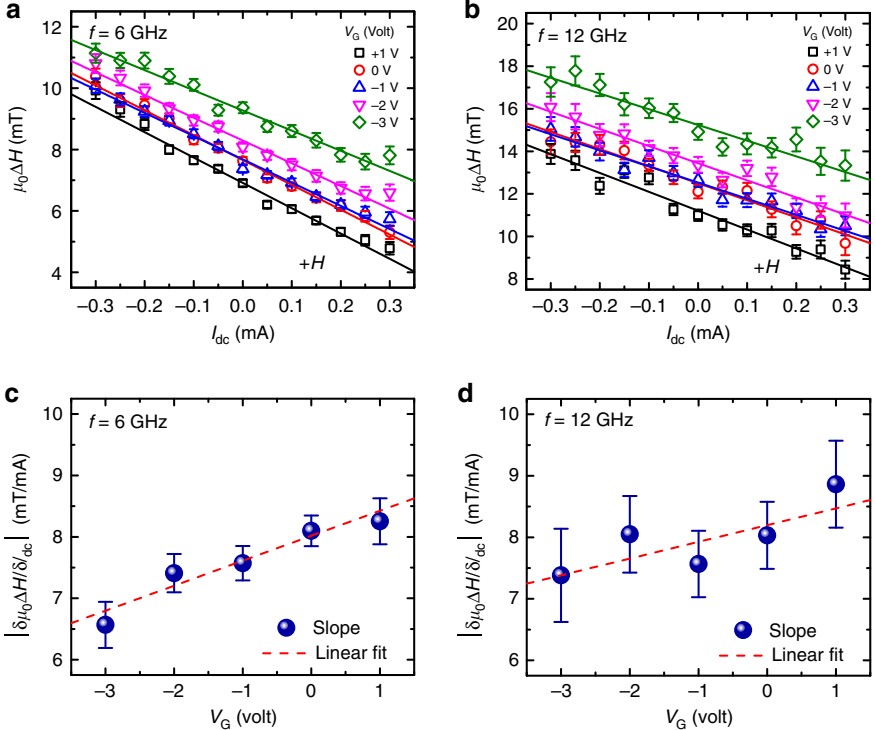

**Fig. 4 Tunability of the dynamical mode localization with gate voltage.** Plots of the current-dependent ST-FMR line-widths, for positive field direction, at two representative frequencies **a** $f = 6$ GHz and **b** $f = 12$ GHz and five different gate voltages. Solid lines are linear fits to the data displaying a minor change in slope with gate voltage. **c**, **d** Variation of extracted slopes with gate voltage indicating the pronounced mode localization at $+1$ V and weaker localization at $-3$ V. Red dashed lines are linear fits to extracted data. All error bars indicate the standard error obtained from fitting the experimental data.

Our ST-FMR results indicate that the large voltage-controlled modulation of the threshold current and auto-oscillation frequency of our SHNOs are produced by two distinct contributions. The former is predominantly caused by the observed strong voltage-controlled change in the effective damping constant, and the latter by the moderate change in effective magnetization. The linear decrease in $\mu_0 M_{\mathrm{eff}}$ with negative gate voltage leads to an effective increase in the FMR frequency as per the Kittel equation[38] for the obliquely magnetized case. This, in turn, increases the auto-oscillation frequency, which essentially mimics the FMR frequency[37].

## Discussion
There are in general two types of VCMA effects reported in the literature to tailor the magnetic properties of FM thin films. One arises due to a redistribution of electronic densities among different $3d$ orbitals in interface FM atoms[20,35,39] or via modification of the Rashba spin-orbit coupling and the Dzyaloshinskii-Moriya interaction (DMI)[24,40]; all these effects have an electronic band structure origin. The other mechanism is of ionic origin, where the voltage-driven migration of $O^{2-}$ ions is exploited to engineer the interfacial oxidation states of the ferromagnetic layer[26,29,41–43]. Unlike engineering the electronic bands at the FM/oxide interface, the ionic mechanism can have a much larger impact on the interface PMA[26,29,42]. Given the observed minor tunability of interface PMA under applied gate voltage, which gives rise to an estimated VCMA value of about $-103.4$ fJ/Vm akin to earlier reported values for this material system[23,44], we believe that the voltage-induced changes in the electronic band structure across the CoFeB/MgO interface have a predominant role in our case.

The damping parameter governs the magnetization dynamics in ferromagnets and therefore critically influences the performance of spintronics devices, such as the auto-oscillation threshold current. While we observed a rather modest VCMA effect of <1%, its substantial impact on the spin-wave mode volume in the nano-constriction region led to the giant 42% modulation of the effective damping. Recently, Nozaki et al. demonstrated a significant enhancement in the VCMA effect, as high as 350 fJ/Vm, via engineering of the FM/oxide interface in an ultrathin Ir-doped Fe/MgO structures with a CoFe termination layer[45]. Employing an ionic mechanism to control the PMA could potentially result in even stronger modulation of the damping and possibly also non-volatile storage of different damping values[46]. Further improvement in the damping modulation should also be possible via optimizing the device layout. For example, by tuning the gate width and constriction opening angle, one can change the ratio between the radiative and core part of the mode, thus control the SHNO damping and frequency tunability.

The demonstrated strong tunability of SHNO threshold current will allow individual oscillator control within long SHNO chains or large two-dimensional SHNO arrays. While a single global drive current is applied to all SHNOs in the chain/array, the voltage applied to each oscillator node determines whether that oscillator will be turned on or stay off. This specific combination of functionalities—i.e., individual on/off control and mutual synchronization—has very recently been suggested based on thermally coupled $VO_2$ relaxation oscillators interacting in two-dimensional arrays[17,18]. This proposed approach will work equally well for two-dimensional SHNO arrays.

Our demonstration of ~50 MHz frequency tunability is also sufficient to cover the frequency tuning used by Romera et al.

(~10 MHz) to optimize the synchronization map for vowel recognition[4]. The SHNO synchronization map demonstrated by Zahedinejad et al. showed detrimental gaps of about 50 MHz, which may hence be possible to close using our demonstrated voltage control[16].

The capability of energy-efficient and individual oscillator control in nano-constriction based SHNO networks has several important advantages over other device layouts for neuromorphic computing applications. First, our device structure is fully integrated without any additional external requirements for injection locking (antenna) and mutual coupling (bonding wires). Second, the nano-constriction based SHNO architecture is highly scalable accommodating as many as 100 partially synchronized oscillators while taking up an area of <1 μm² with the possibility of further downscaling using already demonstrated 20 nm SHNOs[12,16]. Finally, strong non-linear interactions between neighboring oscillators, direct optical access to the active dynamical area, higher operating frequency[10] (3–22 GHz) and a wide locking range (1 GHz)[9,15], both being two orders of magnitude higher than that of vortex STNOs[4], make nano-constriction SHNOs the most viable choice for oscillatory computing. However, in order to truly benefit from these important merits, output power needs to be increased by other means than synchronization. This could, e.g., be done by fabricating a W/CoFeB/MgO/CoFeB magnetic tunnel junction over part of the constriction region by adding a patterned CoFeB layer to the existing structure.

## Methods

**Gated nano-constriction SHNO fabrication.** A trilayer stack of W(5)/(Co$_{0.75}$Fe$_{0.25}$)$_{75}$B$_{25}$(1.7)/MgO(2) (thicknesses in nm) was grown at room temperature on an intrinsic high-resistivity Si substrate ($\rho_{Si} > 10$ kΩ · cm) using an ultra-high vacuum sputtering system. dc and rf sputtering were sequentially employed for the depositions of metallic and insulating layers, respectively. The stack was covered with a 2 nm thin layer of sputtered AlO$_x$ at room temperature to protect the MgO layer from degradation due to exposure to the ambient conditions. The stack was subsequently annealed at 300 °C for 1 h to induce PMA. The stack was then patterned into an array of 4 × 14 μm² rectangular mesas and the nano-constriction SHNO devices with different widths were defined at the center of these mesas by a combination of electron beam lithography and Argon ion beam etching (IBE) using negative electron beam resist as the etching mask. After removing the electron resist, the sample was covered with 15 nm stoichiometric room–temperature sputtered Si$_3$N$_4$ to isolate the gate contacts from the nano-constriction metallic sidewalls. The gates were defined by sputtering a bilayer of Ti (2 nm)/Cu(40 nm) followed by EB lithography using negative electron resist to fabricate 100 nm wide gate pattern on top of nano-constriction. The pattern was then transferred to the Ti/Cu bilayer using IBE technique. After removing the remaining negative resist, the sample went through optical lithography using a positive resist to define vias in Si$_3$N$_4$, giving access to the SHNO metal layers for the contact pads. Finally, the electrical contacts, including two SG-CPWs for microwave and dc measurements were defined by optical lithography and lift-off for a bilayer of Cu(700 nm)/Pt(20 nm).

**Microwave measurements.** Microwave measurements were carried out at room temperature using a custom-built probe station where the sample was mounted at a fixed IP angle of $\phi = 22°$ on an OOP rotatable sample holder between the pole pieces of an electromagnet producing a uniform magnetic field. Using a Keithley 6221 current source, a direct positive electric current, $I_{dc}$, was injected to gated nano-constriction SHNO device through dc port of a high-frequency bias-T under different applied gate voltages sourced from Keithley 2400 sourcemeter. The resulting auto-oscillating signal was then amplified by a low-noise amplifier (LNA) with a gain of ≥32 dB and finally recorded using a spectrum analyzer (SA) from Rhode & Schwarz (10 Hz–40 GHz) comprising a resolution bandwidth of 1 MHz. We measured multiple gated SHNO devices and restricted the maximum current up to 1.4 mA in order to avoid irreversible changes in the output microwave characteristics due to device degradation.

**ST-FMR measurements.** ST-FMR measurements were performed at room temperature on a 150 nm gated nano-constriction SHNO device under the identical conditions employed during auto-oscillation measurements. A radio-frequency (rf) current modulated at 98.76 Hz is supplied through a high-frequency bias-T at a fixed frequency (ranging from 4 to 12 GHz) with an input rf power of $P = -18$ dBm, generating spin-orbit torques as well as Oersted field under an externally applied OOP magnetic field. The resulting torques excite the magnetization

precession in the CoFeB layer, leading to a time-dependent change in the resistance of the device due to the magnetoresistance (MR) of the CoFeB layer[47]. The oscillating MR mixes with the rf current, yielding a dc mixing voltage, $V_{mix}$, and was measured using a lock-in amplifier. All ST-FMR measurements shown in Fig. 3 were carried out by sweeping the applied field oriented at a fixed IP angle of $\phi = 22°$ and OOP angle of $\theta = 60°$ from 550 to 0 mT, while the frequency of the input rf signal is kept fixed. Unlike faster auto-oscillation measurements, the gate breakdown occurred during longer ST-FMR field sweeps at higher positive gate voltage exceeding 1 V. Therefore, we restricted our ST-FMR measurements to +1 V for a fair comparison. To estimate the degree of mode localization in Fig. 4, we injected small dc currents in addition to rf current through dc and rf ports, respectively, of a bias-T. The resonance feature in voltage response from each field sweep was fitted to a sum of one symmetric and one anti-symmetric Lorentzian sharing the same resonance field and linewidth.

The values of $M_{eff}$ were extracted through the fitting by the Kittel equation:

$$f = \frac{\gamma\mu_0}{2\pi}\sqrt{H_{int}(H_{int} + M_{eff}\cos\theta_{int})}, \qquad (1)$$

where $H_{int}$ and $\theta_{int}$ are the internal magnetic field magnitude and out-of-plane angle, respectively, which can be calculated from the magnitude $H_{ex}$ and angle $\theta_{ex}$ of the applied magnetic field as:

$$H_{ex}\cos\theta_{ex} = H_{int}\cos\theta_{int}, \qquad (2)$$

$$H_{ex}\sin\theta_{ex} = (H_{int} + M_{eff})\sin\theta_{int}, \qquad (3)$$

**Micromagnetic simulations.** The micromagnetic simulations were performed using the GPU-accelerated program mumax³[48]. The geometry of a 150-nm nano-constriction width SHNO device was extracted from the CAD model, used for the fabrication, and discretized into 1024 × 1024 × 1 cells with an individual cell size of 4 × 4 × 1.7 nm³. The magnetization dynamics was excited by the rapid ($\sigma = 10$ ps) Gaussian pulse of the current-induced spin-transfer-torque. The in-plane current distribution was modeled in COMSOL. After the pre-relaxation to the linear behavior, the residual oscillations of magnetization, averaged over the under-gate region, was analyzed and fitted by the decay function $\tilde{m} = A\sin(2\pi ft + \phi_0)\exp(-2\pi\Delta ft)$. Here $A$ and $\phi_0$ are the initial amplitude and phase, $f$ is the FMR frequency, and $\Delta f$ is the decay rate. The damping parameter $\alpha$, shown on Fig. 3d was calculated as $\alpha = \frac{\gamma\mu_0}{2\pi}\frac{\Delta f}{f}\frac{dH}{df}$.

In the simulation we used the following parameters: $\mu_0 H_{ext} = 0.4$ T—applied magnetic field, $M_S = 1050$ kA/m—saturation magnetization, $\alpha_0 = 0.009$—Gilbert damping constant[49], $A_{ex} = 19$ pJ/m[50]. The intrinsic PMA value $K_u(V_G = 0) = 645$ kJ/m³ ($t_{CoFeB}K_u = 1096.5$ μJ/m²) with VCMA value of $-2.5$ kJ/m³ per Volt (i.e., $-103.4$ fJ/Vm per area per electric field) were chosen to have a good agreement with the experimentally obtained frequencies by ST-FMR.

## Data availability

The data that support the plots within this paper and other findings of this study are available from the corresponding authors upon reasonable request.

## Code availability

The MATLAB and Mathematica codes used in this study are available from the corresponding authors upon reasonable request. The open-source software package mumax³ used for simulations is available free of charge at https://mumax.github.io/ and http://dynamag.ugent.be/mumax3/ under terms of GNU General Public License version 3.

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

## Acknowledgements

This work was partially supported by the Horizon 2020 research and innovation program (ERC Advanced Grant No. 835068 "TOPSPIN"). This work was also partially supported by the Swedish Research Council (VR) and the Knut and Alice Wallenberg Foundation. Open access funding provided by University of Gothenburg. The work at Tohoku University was supported by JSPS Kakenhi 17H06093 and 19H05622.

## Author contributions

S.F., S.K., and H.O. developed the material stacks. M.Z. designed and fabricated the devices. H.F. carried out all the electrical measurements and data analysis. R.K. and M.D. provided theoretical support with micromagnetic simulations. J.Å. initiated and super-vised the project. All authors contributed to the analysis and interpretation of the results and co-wrote the paper.

## Competing interests

The authors declare no competing interests.
