## [Peer Review File · Nature Communications]

Reviewers' comments:

Reviewer #1 (Remarks to the Author):

The manuscript by Fulara et al. describes an important experimental advance – control of spin-orbit torque oscillator dynamics by gate voltage. Gating of spin-orbit torque oscillators enables control of frequency and damping of individual oscillators in coupled oscillator arrays, which are used for neuromorphic signal processing. The manuscript reveals that voltage-controlled magnetic anisotropy (VCMA) can strongly modulate the effective damping parameter of the oscillator. Comparison of the data to micromagnetic simulations reveals that this giant modulation of damping is due to VCMA-controlled mode confinement, which strongly affects the component of damping due to radiation of magnons into magnetic leads of the nano-constriction oscillator. The manuscript is well written and the conclusions of the paper are strongly supported by the experimental data and micromagnetic simulations. I recommend this paper for publication in Nature Communications if the authors address two comments below:

- Figure 3d shows a convincing agreement between the gate voltage dependence of the measured and simulated values of the oscillator effective damping parameter. For better understanding of the underlying physics, the authors should also present a similar comparison of the measured (e.g. Fig. 2f) and simulated oscillator frequency shift versus gate voltage.

- It is useful to benchmark the magnitude of VCMA (in e.g. fJ/Vm) in the oscillator system studied here to VCMA in other materials systems (e.g. T. Nozaki et al., APL Mater. 8, 011108 (2020); Y.J.Chen et al., Nano Lett. 17, 572 (2017)). Is there room for further enhancement of VCMA tunability of the oscillator frequency and damping if different material combinations are used?

Reviewer #2 (Remarks to the Author):

In this work, Fulara et al demonstrate the control of the properties of spin Hall nano-oscillator by voltage-controlled magnetic anisotropy. Spin Hall nano-oscillators have recently emerged as an excellent building block for neuromorphic computing, and this type of control can provide exciting prospects. In fact, the idea of the current submission was already mentioned by the authors in their recent Nat Nano paper, Ref 14 (“A more useful approach would be direct voltage control of the magnetic properties in the nano-constriction region, such as magnetic anisotropy and/or damping^{33,34}, which could tune the SHNO frequency and possibly also turn them on or off at will.”). This paper, therefore, follows up on that.

Overall, the paper is focused on understanding the physics of the mechanism – neuromorphic computing and applications are only a context. The paper introduces a clear hypothesis for the origin of the effects, which is consistent with multiple ST-FMR and micromagnetic simulations. This analysis is the core contribution of the paper.

Overall, I liked how the paper is constructed. The results are exciting but overall, not oversold. The discussion section adds useful information, and I appreciated how the authors provide leads for enhancing the tunability further. I think that the paper can be inspirational to many.

Speaking of “giant” control of damping seems reasonable to me based on the results (although it will be interesting to see the opinion of more physics-oriented reviewers). On the other hand, I would not

call the frequency tunability “strong” (50MHz, whereas the mean frequency is around 11GHz), so I recommend toning down this particular point throughout the paper (the result itself remains very interesting).

I feel that the paper should discuss more extensively the advantages, and also the drawbacks of this structure with regards to spin torque ones.

So that the paper is more approachable by a diverse audience and that the story stands out, I would recommend moving some of the details of the body text to the Methods section. This would also allow adding a few more sentences so that the basic principle of the structure can be understood without having to study the previous work of the authors.

Reviewer #3 (Remarks to the Author):

The manuscript “Giant voltage control of spin Hall nano-oscillator damping” by H. Fulara, M. Zahedinejad, R. Khymyn, M. Dvornik, S. Fukami, S. Kanai, H. Ohno, & J. Åkerman should be accepted for publication after a minor revision.

The manuscript “Giant voltage control of spin Hall nano-oscillator damping” is devoted to the experimental and numerical/ theoretical study of the damping control in a nano-constriction spin Hall nano-oscillator (SHNO) due to the voltage-controlled magnetic anisotropy effect. The results obtained by the authors evidently show that DC gate voltage (from -2 V to $+2$ V) applied to the nano-oscillator can substantially change its threshold DC current required for the excitation of magnetization dynamics. The authors propose using the discovered effect to control the state of an individual SHNO in the oscillator arrays used for neuromorphic computing, which makes the obtained results to be relevant for the electronics of the future.

The paper is technically sound and clearly written. All the main claims of the manuscript are basically understandably formulated and rather clearly explained.

The paper conclusions are based on evident experimental and numerical results, simple but good theoretical description.

The obtained results are novel; they pave a new way for the implementation of systems for neuromorphic computing.

In general, the article complies with the standards used by the scientific community and is important for the field of applied physics and spintronics.

However, I think several minor corrections should be made:

1. Recently several review papers discussing neuromorphic computing have been published, namely: Sangwan, V.K., Hersam, M.C. Neuromorphic nanoelectronic materials. *Nat. Nanotechnol.* (2020). <https://doi.org/10.1038/s41565-020-0647-z>
Grollier, J., Querlioz, D., Camsari, K.Y. et al. Neuromorphic spintronics. *Nat. Electron.* (2020). <https://doi.org/10.1038/s41928-019-0360-9>

I recommend the authors to cite these papers in the manuscript.

3. In my opinion, there is a small imperfection in the article: several similar abbreviations “DC”, “dc” and “RF”, “rf” are widely used in the manuscript. I suggest the authors to use any but single spelling

of such abbreviations in the whole manuscript, for instance "DC" and "RF" or "dc" and "rf".

Despite of made remarks the reviewed paper has a good scientific quality. Its publication in the Nature Communications is advisable, if the corrections listed above will be made.

Reviewer #4 (Remarks to the Author):

The manuscript reports a large effect of electrostatic gating on the effective damping and the oscillation frequency in spin-Hall nano-oscillators (SHNO) based on ultrathin CoFeB ferromagnets, enabling efficient voltage-controlled tuning and on-off switching of SHNO's oscillation. Surprisingly, the large effect on damping can be explained by a very modest $\sim 1\%$ voltage-induced variation of magnetic anisotropy, which controls the spatial characteristics of the excited dynamical mode, and thus its relaxation due to the coupling to the propagating spin wave modes in the magnetic system. These are nice and somewhat unexpected results that are well-explained in the manuscript and reproduced by the simulations presented by the authors. In my opinion, these results represent a substantial contribution to our understanding of SHNO and the development of efficient and useful spin Hall nanodevices, and are likely to generate a significant interest in the broader spintronics community. Therefore, I recommend the manuscript for publication in Nature Communications, provided that the authors address the following minor suggestions:

i) I would recommend that the wording of the title is changed. One could say that the voltage control is efficient, or perhaps that the effect of voltage on damping is giant, but in the present form the title is confusing. More generally, I recommend that the authors carefully proof-read the manuscript for grammatical errors and confusing phrases, to improve its readability and increase its impact on the research community.

ii) The first sentence of the main text, "High-density and low-power neuromorphic computing requires a large number of mutually synchronized nanoscale spintronic oscillators..." implies that neuromorphic computing can be only implemented with spintronic oscillators, which I do not believe was the meaning of the sentence intended by the authors. It should probably be rephrased.

iii) The initial discussion of the effects of voltage in terms of damping may be somewhat confusing to some readers, because it creates the impression that Gilbert damping is significantly varied, which is not the case. The later descriptions in the manuscript clarify this possible confusion, but I still believe that the relevant terminology needs to be somewhat modified to avoid confusion. As a possible alternative to the term "damping" used by the authors, one can use the term "relaxation rate of the excited dynamical mode". Alternatively, this issue can be addressed by adding a sentence, early on, explicitly specifying that Gilbert damping is not significantly affected.

iv) I do not believe that the use of the term "eigenmode" by the authors in the discussion of the excited dynamical mode is justified, because this dynamical mode is produced by the balance of strongly dissipative processes, and is not really an eigenmode of some Hamiltonian.

Response to Reviews

At the outset, we thank all four Reviewers for their words of appreciation, insightful remarks, and in-depth review of our manuscript. In the following, we present our point-by-point responses to the remarks/suggestions/questions made by each reviewer and highlight the exact changes that we have made to the manuscript. All changes are shown in *blue* in the revised manuscript. We hope that by incorporating these revisions we have adequately addressed the Reviewers' comments and, as a result, we feel that our manuscript has greatly improved.

Reviewer #1 (Remarks to the Author):

Opening remarks: *The manuscript by Fulara et al. describes an important experimental advance – control of spin-orbit torque oscillator dynamics by gate voltage. Gating of spin-orbit torque oscillators enables control of frequency and damping of individual oscillators in coupled oscillator arrays, which are used for neuromorphic signal processing. The manuscript reveals that voltage-controlled magnetic anisotropy (VCMA) can strongly modulate the effective damping parameter of the oscillator. Comparison of the data to micromagnetic simulations reveals that this giant modulation of damping is due to VCMA-controlled mode confinement, which strongly affects the component of damping due to radiation of magnons into magnetic leads of the nano-constriction oscillator. The manuscript is well written, and the conclusions of the paper are strongly supported by the experimental data and micromagnetic simulations.*

Response: We thank the reviewer for appreciating our work. We are also pleased to note that the reviewer describes our work as “...well written, and the conclusions of the paper are strongly supported by the experimental data and micromagnetic simulations.” and recommended our manuscript worthy for publication in Nature Communications.

I recommend this paper for publication in Nature Communications if the authors address two comments below:

Remark #1: *Figure 3d shows a convincing agreement between the gate voltage dependence of the measured and simulated values of the oscillator effective damping parameter. For better understanding of the underlying physics, the authors should also present a similar comparison of the measured (e.g. Fig. 2f) and simulated oscillator frequency shift versus gate voltage.*

Response: We wish to kindly draw the reviewer's attention to the last paragraph before the discussion, where we have explained the observed auto-oscillation frequency shift in terms of experimentally observed voltage-induced changes in $\mu_0 M_{\text{eff}}$ values (See Figure 3(b) in the manuscript) obtained from our ST-FMR measurements. The apparent linear decrease in $\mu_0 M_{\text{eff}}$ with negative gate voltage leads to an increase in FMR frequency in accordance with the Kittel equation and therefore the enhancement of auto-oscillation frequency, which essentially mimics the FMR frequency. We have now rewritten the text in this paragraph for a clearer picture of the underlying physics.

Regarding the micromagnetic simulation of the auto-oscillating frequency as a function of gate voltage, we have found that it depends strongly on many factors that cannot be effectively determined experimentally but have to be assumed. The non-linear frequency shift observed in

auto-oscillations critically influences the frequency tunability with drive current as shown in Fig. 2(e) and depends strongly on the gate voltage, material parameters, geometry of the constriction, local temperature, and most likely the temperature landscape from the strong local variation of the current density. Those parameters, however, have uncertainties and are very much process dependent. For the high supercriticality values, the AO frequency is also strongly governed by non-linear damping and its potential gate voltage dependence, which is also affected by the fabrication process and another non-trivial phenomenon. We hence feel that the strong dependence on simulation parameters makes such micromagnetic simulations less valuable, and potentially misleading, since we simply do not have enough control/knowledge of all parameters. The simulation of the damping, which is also the central result of the paper, is very much different in this regard, since we there take the experimentally observed frequency dependence and can directly estimate the effective total damping without many unknowns.

We think it is very instructive, however, that the voltage dependence of the measured AO frequency closely mimics that of the measured ST-FMR frequency, as shown in the Fig.1 below and mentioned in the manuscript.

Figure 1: Comparison of ST-FMR frequency with auto-oscillation frequency near threshold as a function of gate voltage

“Our ST-FMR results indicate that the large voltage-controlled modulation of the threshold current and auto-oscillation frequency of our SHNOs are produced by two distinct contributions. The former is predominantly caused by the observed strong voltage-controlled change in the effective damping constant, and the latter by the moderate change in effective magnetization. The linear decrease in $\mu_0 M_{\text{eff}}$ with negative gate voltage leads to an effective increase in the FMR frequency in accordance with the Kittel equation Kittel1948 for the obliquely magnetized case. This in turn increases the auto-oscillation frequency, which essentially mimics the FMR frequency.”

Remark #2: *It is useful to benchmark the magnitude of VCMA (in e.g. fJ/Vm) in the oscillator system studied here to VCMA in other materials systems (e.g. T. Nozaki et al., APL Mater. 8, 011108 (2020); Y.J.Chen et al., Nano Lett. 17, 572 (2017)). Is there room for further enhancement of VCMA tunability of the oscillator frequency and damping if different material combinations are used?*

Response: We agree with the reviewer that it is always a good practice to benchmark VCMA values to that in other material systems. We also thank the reviewer for the useful references which are now properly cited in the manuscript. With regards to the question of the reviewer, we do believe there is a room for further enhancement of the VCMA tunability using different material combinations via engineering of the FM/oxide interface, for example with the doping/alloying using a heavy metal like Ir, as reported in the suggested references. Recently, a large VCMA magnetoionic effect (F. Xue et al., APL Materials 7, 101112 (2019)) was observed in the identical CoFeB/MgO/SiO₂ double oxide structure, and we have mentioned this point, as also underlined by the reviewer 2, in the discussion section. Also, while the increasing of the VCMA is important, further optimization of the device layout in our case can improve the performance even more. For example, by tuning the gate width and constriction opening angle one can change the ratio between radiative and core part of the mode, thus control the oscillator damping and frequency tunability. Now we have re-written the text in the discussion section while accommodating new references suggested by the reviewer as:

“.....Given the observed minor tunability of interface PMA under applied gate voltage, which gives rise to an estimated VCMA value of about -103.4 fJ/Vm akin to earlier reported values for this material system\cite{wang2012natmat,chen2017nanl}, we believe that the voltage-induced changes in the electronic band structure across the CoFeB/MgO interface have a predominant role in our case.

The damping parameter governs the magnetization dynamics in ferromagnets and therefore critically influences the performance of spintronics devices, such as the auto-oscillation threshold current. While we observed a rather modest VCMA effect of less than 1 %, its substantial impact on the spin-wave mode volume in the nano-constriction region led to the giant 42 % modulation of the effective damping. Recently, Nozaki et al. demonstrated a significant enhancement in the VCMA effect, as high as 350 fJ/V.m, via engineering of the FM/oxide interface in an ultrathin Ir-doped Fe/MgO structures with a CoFe termination layer.\cite{nozaki2020aplm} Employing an ionic mechanism to control the PMA could potentially result in even stronger modulation of the damping and possibly also non-volatile storage of different damping values.\cite{xue2019aplm} Further improvement in the damping modulation should also be possible via optimizing the device layout. For example, by tuning the gate width and constriction opening angle, one can change the ratio between the radiative and core part of the mode, thus control the SHNO damping and frequency tunability.”

Reviewer #2 (Remarks to the Author):

Opening remarks: *In this work, Fulara et al demonstrate the control of the properties of spin Hall nano-oscillator by voltage-controlled magnetic anisotropy. Spin Hall nano-oscillators have recently emerged as an excellent building block for neuromorphic computing, and this type of control can provide exciting prospects. In fact, the idea of the current submission was already mentioned by the authors in their recent Nat Nano paper, Ref 14 (“A more useful approach would be direct voltage control of the magnetic properties in the nano-constriction region, such as magnetic anisotropy and/or damping^{33,34}, which could tune the SHNO frequency and possibly also turn them on or off at will.”). This paper, therefore, follows up on that.*

Overall, the paper is focused on understanding the physics of the mechanism – neuromorphic computing and applications are only a context. The paper introduces a clear hypothesis for the origin of the effects, which is consistent with multiple ST-FMR and micromagnetic simulations. This analysis is the core contribution of the paper.

Overall, I liked how the paper is constructed. The results are exciting but overall, not oversold. The discussion section adds useful information, and I appreciated how the authors provide leads for enhancing the tunability further. I think that the paper can be inspirational to many.

Response: We are extremely pleased to note that the reviewer has appreciated the motivation part and has underlined the main idea behind the present work. We also thank the reviewer for highlighting the important points and describing our work as “... a clear hypothesis for the origin of the effects, which is consistent with multiple ST-FMR and micromagnetic simulations. This analysis is the core contribution of the paper.” and “... the paper can be inspirational to many.”

Remark #1: *Speaking of “giant” control of damping seems reasonable to me based on the results (although it will be interesting to see the opinion of more physics-oriented reviewers). On the other hand, I would not call the frequency tunability “strong” (50MHz, whereas the mean frequency is around 11GHz), so I recommend toning down this particular point throughout the paper (the result itself remains very interesting).*

Response: We agree with the reviewer and therefore have replaced the word “strong” by “robust” throughout the paper.

Remark #2: *I feel that the paper should discuss more extensively the advantages, and also the drawbacks of this structure with regards to spin torque ones.*

So that the paper is more approachable by a diverse audience and that the story stands out, I would recommend moving some of the details of the body text to the Methods section. This would also allow adding a few more sentences so that the basic principle of the structure can be understood without having to study the previous work of the authors.

Response: We are thankful to the reviewer for this insightful remark and now added a sentence in the introduction section describing the basic principle and a paragraph in the discussion section highlighting the advantages and future challenges.

Introduction:

“SHNOs take advantage of a pure spin current, produced by the spin Hall effect in a non-magnetic heavy-metal layer, to excite the local magnetization of a magnetic thin film into steady state auto-oscillating precession at microwave frequencies.”

Discussion:

“The capability of energy-efficient and individual oscillator control in nano-constriction based SHNO networks has several important advantages over other device layouts for neuromorphic computing applications. First, our device structure is fully integrated without any additional external requirements for injection locking (antenna) and mutual coupling (bonding wires). Second, the nano-constriction based SHNO architecture is highly scalable accommodating as many as 100 partially synchronized oscillators while taking up an area of less than $1 \mu\text{m}^2$ with the possibility of further downscaling using already demonstrated 20 nm SHNOs\cite{Zahedinejad2020natnano,durrenfeld2017nanoscale}. Finally, strong non-linear interactions between neighboring oscillators, direct optical access to the active dynamical area, higher operating frequency\cite{Zahedinejad2018apl} (3 to 22 GHz) and a wide locking range (1 GHz),\cite{Awad2016natphys,Fulara2019SciAdv} both being two orders of magnitude higher than that of vortex STNOs\cite{romera2018nature}, make nano-constriction SHNOs the most viable choice for oscillatory computing. However, in order to truly benefit from these important merits, output power needs to be increased by other means than synchronization. This could \emph{e.g.}~be done by fabricating a W/CoFeB/MgO/CoFeB magnetic tunnel junction over part of the constriction region by adding a patterned CoFeB layer to the existing structure.”

Reviewer #3 (Remarks to the Author):

Opening remarks: *The manuscript “Giant voltage control of spin Hall nano-oscillator damping” by H. Fulara, M. Zahedinejad, R. Khymyn, M. Dvornik, S. Fukami, S. Kanai, H. Ohno, & J. Åkerman should be accepted for publication after a minor revision.*

The manuscript “Giant voltage control of spin Hall nano-oscillator damping” is devoted to the experimental and numerical/ theoretical study of the damping control in a nano-constriction spin Hall nano-oscillator (SHNO) due to the voltage-controlled magnetic anisotropy effect. The results obtained by the authors evidently show that DC gate voltage (from -2 V to $+2$ V) applied to the nano-oscillator can substantially change its threshold DC current required for the excitation of magnetization dynamics. The authors propose using the discovered effect to control the state of an individual SHNO in the oscillator arrays used for neuromorphic computing, which makes the obtained results to be relevant for the electronics of the future.

The paper is technically sound and clearly written. All the main claims of the manuscript are basically understandably formulated and rather clearly explained. The paper conclusions are based on evident experimental and numerical results, simple but good theoretical description. The obtained results are novel; they pave a new way for the implementation of systems for neuromorphic computing. In general, the article complies with the standards used by the scientific community and is important for the field of applied physics and spintronics.

Response: *We are grateful to the reviewer for appreciating our work in length. We are particularly pleased to note that the reviewer has rated our work as “... technically sound and clearly written. All the main claims of the manuscript are basically understandably formulated and rather clearly explained. The paper conclusions are based on evident experimental and numerical results, simple but good theoretical description. The obtained results are novel; they pave a new way for the implementation of systems for neuromorphic computing. In general, the article complies with the standards used by the scientific community and is important for the field of applied physics and spintronics.”*

However, I think several minor corrections should be made:

Remark #1: *Recently several review papers discussing neuromorphic computing have been published, namely:*

Sangwan, V.K., Hersam, M.C. Neuromorphic nanoelectronic materials. Nat. Nanotechnol. (2020). <https://doi.org/10.1038/s41565-020-0647-z>

Grolier, J., Querlioz, D., Camsari, K.Y. et al. Neuromorphic spintronics. Nat. Electron. (2020). <https://doi.org/10.1038/s41928-019-0360-9>

I recommend the authors to cite these papers in the manuscript.

Response: *We are thankful to the reviewer for suggesting the important recent references for our work and they have now been incorporated in the introduction of our manuscript.*

Remark #2: *In my opinion, there is a small imperfection in the article: several similar abbreviations “DC”, “dc” and “RF”, “rf” are widely used in the manuscript. I suggest the authors to use any but single spelling of such abbreviations in the whole manuscript, for instance “DC” and “RF” or “dc” and “rf”.*

Despite of made remarks the reviewed paper has a good scientific quality. Its publication in the Nature Communications is advisable, if the corrections listed above will be made.

Response: We again thank the reviewer for highlighting the inconsistency of using abbreviations, which have now been corrected in the manuscript.

Reviewer #4 (Remarks to the Author):

Opening remarks: The manuscript reports a large effect of electrostatic gating on the effective damping and the oscillation frequency in spin-Hall nano-oscillators (SHNO) based on ultrathin CoFeB ferromagnets, enabling efficient voltage-controlled tuning and on-off switching of SHNO's oscillation. Surprisingly, the large effect on damping can be explained by a very modest ~1% voltage-induced variation of magnetic anisotropy, which controls the spatial characteristics of the excited dynamical mode, and thus its relaxation due to the coupling to the propagating spin wave modes in the magnetic system. These are nice and somewhat unexpected results that are well-explained in the manuscript and reproduced by the simulations presented by the authors. In my opinion, these results represent a substantial contribution to our understanding of SHNO and the development of efficient and useful spin Hall nanodevices and are likely to generate a significant interest in the broader spintronics community.

Response: We thank the reviewer for the in-depth review, expert remarks and are particularly pleased to note that the reviewer finds our results “*a substantial contribution to our understanding of SHNO and the development of efficient and useful spin Hall nanodevices and are likely to generate a significant interest in the broader spintronics community.*”

Therefore, I recommend the manuscript for publication in Nature Communications, provided that the authors address the following minor suggestions:

Remark #1: I would recommend that the wording of the title is changed. One could say that the voltage control is efficient, or perhaps that the effect of voltage on damping is giant, but in the present form the title is confusing. More generally, I recommend that the authors carefully proof-read the manuscript for grammatical errors and confusing phrases, to improve its readability and increase its impact on the research community.

Response: We have changed the title to “**Giant voltage-controlled modulation of spin Hall nano-oscillator damping**”. We have also gone over our manuscript in detail and hope that we have now caught all grammatical issues.

Remark #2: The first sentence of the main text, “High-density and low-power neuromorphic computing requires a large number of mutually synchronized nanoscale spintronic oscillators...” implies that neuromorphic computing can be only implemented with spintronic oscillators, which I do not believe was the meaning of the sentence intended by the authors. It should probably be rephrased.

Response: We are grateful to the reviewer for bringing this to our notice. We have now re-written the sentence for better clarity as under:

“High-density arrays of mutually coupled nanoscale spintronic oscillators have the potential to closely mimic the behavior of the non-linear oscillatory neural networks of the human brain and thus open the way for efficient high-speed and low-power neuromorphic computing and signal processing\cite{grollier2016ieee,grollier2020natelc,sangwan2020natnano}”

Remark #3: The initial discussion of the effects of voltage in terms of damping may be somewhat confusing to some readers, because it creates the impression that Gilbert damping is significantly varied, which is not the case. The later descriptions in the manuscript clarify this possible confusion, but I still believe that the relevant terminology needs to be somewhat modified to avoid confusion. As a possible alternative to the term “damping” used by the authors, one can use the term “relaxation rate of the excited dynamical mode”. Alternatively, this issue can be addressed by adding a sentence, early on, explicitly specifying that Gilbert damping is not significantly affected.

Response: We understand the reviewer’s concern about the use of appropriate terminology to avoid confusion with Gilbert damping. In fact, we refrained from using the word “Gilbert” in the manuscript, instead used the word “effective” to this effect. What we meant by damping in the manuscript is the total effective damping, which includes “intrinsic” contribution, “interfacial” contribution, and the dominant “radiative” part in our case. Now to avoid any confusion, we have added a term “total effective” before damping in the introduction section. We have also included the term “radiation of magnons into the magnetic leads” in the introductory text and “relaxation rate of the excited dynamical mode” in the main text to avoid any further confusion.

Remark #4: I do not believe that the use of the term “eigenmode” by the authors in the discussion of the excited dynamical mode is justified, because this dynamical mode is produced by the balance of strongly dissipative processes, and is not really an eigenmode of some Hamiltonian.

Response: We agree with reviewer's concern about the choice of the term “eigenmode” and therefore now changed “eigenmode” to “linear-mode”.

REVIEWERS' COMMENTS:

Reviewer #1 (Remarks to the Author):

The authors have adequately addressed all comments of all four reviewers and modified the manuscript accordingly. In my opinion, the manuscript can be published in Nature Communications without any additional revisions.

Reviewer #2 (Remarks to the Author):

The authors have addressed my comments. I recommend the publication of the manuscript.

Reviewer #3 (Remarks to the Author):

The authors took into account all the referee's comments. The authors' answers to these comments are constructive and professional. They are written in a friendly tone. The necessary changes have been made to the manuscript.

Therefore, in my humble opinion, the manuscript in its present form CAN BE published in the Nature Communications.

Reviewer #4 (Remarks to the Author):

The authors have thoroughly addressed the comments of of all 4 reviewers in the manuscript text and in their responses. The manuscript is suitable for publication in the present form.